# Prognostic Value of Ultra-Short Heart Rate Variability Measures Obtained from Electrocardiogram Recordings of Hospitalized Patients Diagnosed with Non-ST-Elevation Myocardial Infarction

**DOI:** 10.3390/jcm13237255

**Published:** 2024-11-28

**Authors:** Maya Reshef, Shay Perek, Tamer Odeh, Khalil Hamati, Ayelet Raz-Pasteur

**Affiliations:** 1Internal Medicine “A” Department, Rambam Medical Health Care Campus, Haifa 3109601, Israel; s_perek@rambam.health.gov.il (S.P.); t_odeh@rambam.health.gov.il (T.O.); khalilh.hamati@gmail.com (K.H.); 2Emergency Medicine Department, Rambam Medical Health Care Campus, Haifa 3109601, Israel; 3The Ruth and Bruce Rappaport Faculty of Medicine, Technion-Israel Institute of Technology, Haifa 3525422, Israel

**Keywords:** ultra-short heart rate variability, non-ST-elevation myocardial infraction, ventricular tachycardia, ventricular fibrillation, mortality

## Abstract

**Background:** Myocardial infarction (MI) is a common emergency with high rates of morbidity and mortality. Current risk stratification scores for non-ST-elevation MI (NSTEMI) use subjective or delayed information. Heart rate variability was shown to correlate with prognosis following MI. This study aimed to evaluate ultra-short heart rate variability (usHRV) as a prognostic factor in NSTEMI patients. **Methods**: A retrospective analysis was performed on 183 NSTEMI patients admitted to Rambam Health Care Campus in 2014. usHRV measures, including the standard deviation of normal-to-normal intervals (SDNN) and root mean square of successive differences (RMSSD), were calculated. Logistic regression assessed whether clinical, laboratory, or usHRV parameters predicted severe in-hospital complications like heart failure (HF), atrial flutter/fibrillation (AFL/AF), ventricular tachycardia/fibrillation (VT/VF), and atrioventricular block (AVB). Both Cox and logistic regression were used for survival analysis. **Results**: Of 183 patients (71.6% male, mean age 67.1), 35 (19%) died within 2 years. In-hospital complications included 39 cases (21.3%) of HF, 3 cases (1.6%) of VT/VF, and 9 cases (4.9%) of AVB. Lower usHRV was significantly associated with higher mortality at 2 years and showed marginal significance at 90 days and 1 year. Increased usHRV was linked to a higher risk of in-hospital ventricular arrhythmia (VT/VF). **Conclusions**: Overall, this study is in agreement with previous research, showing a correlation between low usHRV and a higher mortality risk. However, the association between usHRV and the risk of VT/VF demands further investigation. More expansive prospective studies are needed to strengthen the observed associations.

## 1. Introduction

Non-ST-elevation myocardial infarction (NSTEMI) is an acute coronary syndrome (ACS) that continues to carry substantial rates of both morbidity and mortality [1,2]. The population of patients diagnosed with NSTEMI displays heterogeneity in prognosis, with considerable variability in risks of mortality and reinfarction [3,4]. Recent population studies depict an increase in the relative incidence of NSTEMI, rising from 14.2% in 1999 to 59.1% in 2006, with a corresponding decrease in ST-elevation myocardial infarction (STEMI) incidence [5]. Proposed explanations for these changes include the implementation of high-sensitivity troponin (hsTn) assays in ACS diagnosis, general population aging, and increases in the prevalence of certain risk factors for coronary artery disease (CAD) [6].

Multiple risk stratification models were developed to address the high heterogeneity among NSTEMI patients and their predicted risk of death or MACE [7,8]. The Thrombolysis In Myocardial Infarction (TIMI) [9] risk score was able to predict the short-term risk of MACE within 14 days, while the Global Registry of Acute Coronary Events (GRACE) [10] risk score is predictive of mortality during hospitalization and within 6 months to 1 year. This study aims to predict longer-term MACE up to 2 years [11] and provide additional insight into the determinants of prognosis. In addition, TIMI and GRACE scores both require the evaluation of clinical indicators of ACS to determine risk. This study aims to propose a tool that relies on a 10 s ECG recording on admission for risk stratification. Such a tool will likely simplify risk assessment and facilitate timely decision making, which is especially important for high-risk patients.

In recent years, heart rate variability (HRV) has emerged as a promising prognostic tool used to infer the relative contribution of parasympathetic and sympathetic divisions of the ANS, among other regulatory inputs [12,13,14]. Parasympathetic (vagal) activity lowers HR and increases HRV, while the opposite is true for sympathetic activity. Reduced HRV was shown to predict adverse prognosis in a wide range of clinical conditions including sepsis [15,16], chronic kidney disease [17], myocarditis [18], and cancer [19]. Reduced HRV was also demonstrated to predict an increased mortality risk both in the elderly population and in the general population [20,21]. In 1978, a study in MI patients observed that the presence of sinus arrhythmia, which leads to variations in heart rate, was associated with a lower in-hospital mortality rate [22]. Later studies showed that low HRV following MI is associated with an increased risk of all-cause mortality [23] and cardiac mortality [24,25], and specifically of mortality from heart failure (HF) [25,26] and sudden cardiac death [27].

Traditionally, studies used HRV measures extracted from 24 h Holter recordings. Yet multiple studies indicate that shorter electrocardiogram (ECG) recordings may be used to calculate some of the commonly used HRV measures, such as standard deviation of normal-to-normal intervals (SDNN) and root mean square of successive differences (RMSSD) [22,28,29,30]. One study tested the validity of ultra-short HRV (usHRV) measurements and found that 10 s RMSSD and SDNN are reliable proxies for those measured from longer recordings, correlating moderately to strongly with longer recordings [31]. This implies that the standard admission 10 s ECG itself may yield valuable prognostic information that can help identify patients at need of more intensive care or an early-invasive treatment strategy. One study demonstrated that reduced usHRV on admission ECG was associated with increased mortality at 1 and 6 months and at 1 and 2 years following STEMI [32]. However, the prognostic value of usHRV in NSTEMI patients has not yet been fully characterized. In light of previous evidence, we hypothesize that reduced usHRV may also serve as an independent predictor of increased short- and long-term mortality in patients following NSTEMI.

The main aim of this study was to explore potential correlations between usHRV variables in patients presenting with NSTEMI and mortality, as well as major adverse cardiovascular events, and characterize them. The findings may provide prognostic insights, offering a basis for future risk stratification and management decision making in NSTEMI patients.

## 2. Materials and Methods

### 2.1. Study Design and Patient Population

In this retrospective, observational, single-center cohort study, we sequentially enrolled all patients admitted to Rambam Health Care Campus (RHCC; Haifa, Israel) between 1 January and 31 December 2014 with a diagnosis of NSTEMI. We included all NSTEMI patients for whom a 10 s resting ECG was recorded on admission. We excluded patients with recordings that were not recovered or were shorter, as well as those with recordings of low resolution, for which usHRV data could not be obtained. We also excluded patients whose ECG recordings showed evidence of arrhythmia, precluding usHRV measurement.

### 2.2. Data Collection

All emergency department (ED) visits and hospital discharge letters from the study period were screened for a diagnosis of NSTEMI, utilizing the MDClone version 6.3 (Beer-Sheva, Israel) computer software. This study included only patients who had undergone a 10 s resting ECG in the ED. In addition, all medical records were reviewed in order to identify and exclude potentially ineligible patients. Patients’ medical history (including NSTEMI risk factors), ED vital signs, and laboratory results (i.e., complete blood count, chemistry panel, and cardiac enzymes), were collected. Patient data regarding primary and secondary endpoints of this study were collected either using the MDClone software or manually from patients’ medical records. HF exacerbations were defined as both typical clinical manifestations along with a reduction in left ventricular ejection fraction. Ventricular and supra-ventricular arrhythmias were diagnosed based on either admission ECG or 24 h Holter examinations or based on the physician report in the hospital discharge note. Furthermore, all included patients’ charts were reviewed in order to identify disease-specific complications and date of death.

This study was approved by the Helsinki committee at RHCC, Helsinki approval number 0603-16-RMB. Since all patient data were collected retrospectively, informed consent was not required. The datasets analyzed during this current study are available in the FigShare repository, using the following hyperlink: https://doi.org/10.6084/m9.figshare.26905000.v1 (accessed on 2 September 2024).

### 2.3. ECG Analysis and Computation of usHRV Indices

All patients arriving at RHCC ED with a suspected acute cardiovascular disease underwent immediate ECG (LAN Green model, Norav Medical, Yokne’am, Israel) while lying motionless in the supine position for at least 30 s, within the ED non-ambulatory wing. Patients remained still during the exam preparation for more than 1 min. ECG electrodes were placed in anatomical positions according to standard procedure. ECG capture required patients to remain very still for an additional 10 s. In case the ECG quality was subpar, another study was immediately performed. Finally, if the ECG study, which had been recorded during 2014, was partially complete or of low quality, the patient was excluded during our analysis. An ECG viewing program was used to visualize the resting ECG files (Resting ECG version 5.62, Norav Medical). PR interval duration and QRS interval duration were automatically measured, and QT interval duration was calculated based on the Bazzett equation. Later, ECG files were analyzed with a custom version of the HRV analysis software able to import and analyze 10 s long recordings (HRV version 5.62, Norav Medical). These programs allowed for the automatic computation of usHRV parameters. In addition, ECGs were manually checked, and recordings with disturbances (e.g., excessive noise, sudden baseline instability, or low resolution) were excluded. ECGs that contained excessive premature ventricular or supraventricular activity (e.g., atrial fibrillation, atrial flutter, atrial tachycardiac), as well as advanced atrio-ventricular conduction abnormalities (e.g., second degree atrioventricular blocks, complete heart block, or other high-degree conduction abnormalities), were also excluded. Linear time-domain usHRV variables (e.g., standard deviation of RR intervals [SDNN] and root mean square of successive differences [RMSSD]) were mainly used in our study in light of their high agreement with longer-term recordings.

### 2.4. Statistical Analysis

The study database was analyzed with R software (version 4.0.3, The R Foundation for Statistical Computing, Vienna, Austria). Continuous variables are presented in means and standard deviation (SD), unless otherwise specified. Categorical variables are presented in absolute numbers and percentages. Comparisons between groups were performed with Student’s *t*-test or the Mann–Whitney U test for continuous variables and with Fisher’s exact test for categorical variables. Correlations between variables and Boolean outcomes were examined with logistic regressions (LRs) and presented as an odds ratio (OR) with 95% confidence interval (CI) and *p*-values. Variables found to have statistical significance (*p*-value < 0.05) in univariate analyses were introduced into a multivariate LR model in a backward stepwise fashion. Multivariate model accuracy is presented with receiver operating characteristic (ROC) curves, including the area under the curve (AUC), with 95% CIs based on the bootstrapping method, as well as by evaluating the Hosmer–Lameshow goodness-of-fit (HLGOF) and overall model *p*-value. UsHRV parameters were assessed as both continuous and Boolean variables (e.g., smaller than values corresponding to quartile 1 [Q1; 25th percentile]). Cox regression survival analysis was also performed for usHRV measures.

### 2.5. Study Endpoints

The primary endpoint of the study was to determine the value of usHRV in predicting short- and long-term mortality following NSTEMI. Secondary endpoints were to assess the value of usHRV in predicting the duration of index hospitalization, as well as in-hospital adverse events, including new-onset heart failure (HF) or exacerbation in preexisting HF, supraventricular tachycardia (SVT) or atrial fibrillation (AF), either sustained or non-sustained ventricular tachycardia (VT) or fibrillation (VF), and 2nd- or 3rd-degree atrioventricular blocks (AVBs).

## 3. Results

### 3.1. Patient Population and Baseline Characteristics

A total of 282 patients were admitted to RHCC throughout 2014 with a diagnosis of NSTEMI and were retrospectively enrolled in this study. We excluded 20 patients whose ECG recordings could not be found or were too short for evaluation and 17 patients for whom usHRV data could not be obtained. We then excluded 62 patients due to arrhythmia on their ECG. Thus, 183 patients were ultimately enrolled in the study (Figure 1).

The baseline characteristics of the patients are presented according to those who did and did not survive 2 years following NSTEMI diagnosis (Table 1). The mean age of patients was 67.1 years (SD ± 14.4, range 35.2–99.5), and 71.6% were male. Of the patients enrolled, 35 (19%) died within 2 years of NSTEMI diagnosis. On average, these patients were older (79.0 ± 11.0 vs. 64.3 ± 13.7 years) and more likely to have previously been diagnosed with HF (32.4% vs. 11.0%). Baseline creatinine and blood urea nitrogen (BUN) levels were higher in patients who did not survive than in survivors. Survivors, on the other hand, presented relatively slower heart rates (80.2 vs. 87.8 bpm), which is in line with the higher median SDNN displayed in this group (13.12 vs. 10.28 ms).

### 3.2. All-Cause Mortality Within 30 and 90 Days, and 1 and 2 Years of NSTEMI Diagnosis

We performed logistic regression analysis to determine whether usHRV measures are associated with all-cause mortality at 30 and 90 days, and 1 and 2 years following NSTEMI (Table 2). A two-year survival analysis revealed that low SDNN significantly increases the risk of long-term mortality following NSTEMI (OR: 0.956, *p*-value: 0.045). Marginally significant associations were found between low SDNN and an increased mortality risk at both 90 days and 1 year following NSTEMI (OR: 0.901, *p*-value: 0.061 and OR: 0.955, *p*-value: 0.078, respectively). Survival analysis within 30 days following NSTEMI did not reveal significant associations between usHRV and mortality risk.

#### All-Cause Mortality over Time

To characterize the correlation between usHRV measures and survival, we also performed Cox proportional hazards survival analysis, which did not identify significant interactions between any of the usHRV measures and survival time. SDNN: Hazard ratio (HR)—0.992 (CI: 0.978–1.006), *p*-value 0.259; RMSSD: HR—0.998 (CI: 0.990–1.007), *p*-value 0.802; HTI: HR—0.882 (CI: 0.743–1.048), *p*-value 0.154.

### 3.3. In-Hospital New-Onset HF or Exacerbation in Preexisting HF

A total of 39 patients (21.3%) developed new-onset HF or exacerbation in preexisting HF. Univariable logistic regression analysis was performed to assess correlations between usHRV and the development of in-hospital, new-onset HF or exacerbation in preexisting HF (Table A1). No associations were found between usHRV measures and in-hospital HF.

### 3.4. In-Hospital New-Onset VF or VT

A total of three patients (1.6%) developed new-onset VF or VT. Univariable logistic regression analysis was performed to determine whether usHRV is associated with the development of new-onset VF or VT during the index hospitalization with NSTEMI (Table 3). The results exhibit significant associations between both increased SDNN (OR: 1.025, *p*-value: 0.020) and increased RMSSD (OR: 1.018, *p*-value: 0.016) and an increased risk of developing new-onset VF or VT in-hospital with NSTEMI, although it should be noted that only three patients developed this early adverse event.

Multivariable logistic regression included WBCs and SDNN in the prediction model for the early development of ventricular arrhythmias during NSTEMI hospitalization (Table 4). Receiver-operating characteristic (ROC) analysis had good predictive performance, with an area under the curve (AUC) of 0.874 (95% CI 0.674–1.000), as presented in Figure 2.

### 3.5. In-Hospital New-Onset SVT, AF, or AFL

A total of 11 patients (6%) developed new-onset SVT, AF, or AFL. Univariable logistic regression analysis was performed to determine the predictive value of usHRV for developing in-hospital SVT, AF, or AFL (Table A2). No associations were observed between usHRV and this risk.

### 3.6. In-Hospital New-Onset AVB

A total of nine patients (4.9%) developed new-onset AVB. Univariable logistic regression analysis was performed to determine the predictive value of usHRV for the risk of developing in-hospital AVB (Table A3). The results did not present significant associations between usHRV and this risk.

## 4. Discussion

This retrospective study demonstrates the predictive value of time-domain usHRV measures for long-term mortality in NSTEMI patients. We found a significant association between low SDNN and increased all-cause mortality risk at two years. Notably, we also found an association between high usHRV measures and an increased risk of in-hospital VF/VT in both univariable and multivariable analyses.

Low SDNN was significantly associated with an increased 2-year mortality risk in patients following NSTEMI. This is in line with our hypothesis and with multiple previous studies that demonstrated similar associations [25,32]. According to our results, every 1 ms increase in SDNN is associated with a 4.4% reduction in 2-year mortality risk, which can be clinically important when used in combination with established risk scores. With that, it should be noted that an OR of 0.956 implies that the effect size of low SDNN on mortality is relatively small and that the 95% CI upper bound approaching one suggests some uncertainty in the precise effect size, which may be slightly weaker or stronger. This points to the need for larger-scale studies for a more accurate estimation of effect size and validation.

This study identified a significant association between high usHRV and the increased risk of in-hospital VF/VT in both univariable analyses, for SDNN and RMSSD, and multivariable analysis for SDNN. This finding contrasts with our hypothesis and with the widely accepted view that high HRV reflects cardiovascular health. The results of this study were obtained, given the low number of VF/VT events (*n* = 3), which explains the wide 95% CI observed in both univariable and multivariable analyses and indicates a substantial degree of uncertainty. Controversy still exists regarding the nature of HRV alterations and their relation to the development of ventricular tachyarrhythmias in cardiac patients. While reduced HRV is most commonly associated with increased risk of ventricular arrhythmias following MI [33,34,35], high HRV was also linked with adverse outcomes in several pathological conditions. In the elderly population, high HRV was associated with increased risks of all-cause mortality and fatal arrhythmias [21,36]. In other contexts, high usHRV predicted increased mortality in patients with bacterial pneumonia and infective endocarditis [37,38], with increased probability of a new arrhythmia in patients with infective endocarditis [38]. These studies provide a basis to support the correlation between high usHRV and an increased risk of VF/VT. Nonetheless, this correlation is of uncertain validity and warrants further investigation into the pathophysiological mechanisms behind high HRV measures and adverse outcomes following NSTEMI.

In general, short-term HRV measures reflect more strongly on parasympathetic function than on sympathetic activity, as vagal activity is mediated by acetylcholine, which acts more rapidly at the sino-atrial node (SAN) compared with sympathetic modulation via norepinephrine [12]. This suggests that low usHRV is more likely to reflect vagal dysfunction. Increased sympathetic activity was shown to increase electrical instability, lowering the ventricular fibrillation threshold [33,39,40,41]. However, an increased risk of VF/VT may also be considered in the context of high usHRV. Although such a correlation has not been clearly defined, high usHRV can also reflect irregular RR-interval [42]. SAN dysfunction, and sick sinus syndrome in particular, may arise from structural, ischemic, or fibrotic damage to the SAN and surrounding tissues, which can lead to the development of highly variable, irregular RR intervals, similar to those observed in sick sinus syndrome [43,44]. As such, a correlation between high HRV and an increased risk of ventricular arrhythmia may be explained by SAN dysfunction, which may result from acute ischemia [21,36]. However, given the limited validity of our findings, further studies are warranted to explore this hypothesis and validate the results observed in this study.

The use of HRV measures was previously shown to be a valuable important factor in the prognosis of NSTEMI. In this study, we explore the potential role of usHRV as a novel tool for predicting long-term outcomes in NSTEMI patients. Our results suggest that usHRV may provide valuable prognostic information, particularly in identifying patients at a high risk of long-term mortality. The incorporation of usHRV measurement is highly practical, as it only requires the implementation of relevant analysis software to provide real-time measures for use in risk stratification models. The introduction of usHRV measures into the clinical workflow could help identify high-risk NSTEMI patients who will be more likely to benefit from hospitalization in either intensive-care units or cardiology departments, as well as guide the need for urgent diagnostic and therapeutic intervention. Nonetheless, leveraging the potential of usHRV in clinical practice will require further investigation, including larger-scale and multicenter prospective trials and more in-depth, mechanistic insights.

## 5. Limitations

There were several important limitations in this study. First, the low number of individual outcome events substantially limited the statistical power of our results, particularly when discussing the analysis of rare events such as VF/VT, which led to the high confidence intervals presented. This highlights the need for larger, multi-centric trials, to increase the numbers of patients that develop each of the important outcome events discussed. Second, the identification of complications was based on manual scanning of individual patient files for the documentation of such events. As such, patients were not under targeted follow-up, and, therefore, it is possible that not all complications were identified. Third, we were unable to determine the cause of death for all study patients, which led to the primary outcome being all-cause mortality, rather than cardiovascular mortality. Fourth, the retrospective design of the study is inherently limited by its reliance on pre-existing data, which can be incomplete, inconsistent, or prone to biases. Fifth, a single-center design limits the generalizability of findings, as the patient population and care practices may not reflect other hospitals or regions. A single-center study may be more vulnerable to biases specific to that institution, such as selection bias or local practice patterns. We realize that the reliability of results obtained under the conditions of this study may be of limited value and therefore advise readers to interpret these results with caution.

## Figures and Tables

**Figure 1 jcm-13-07255-f001:**
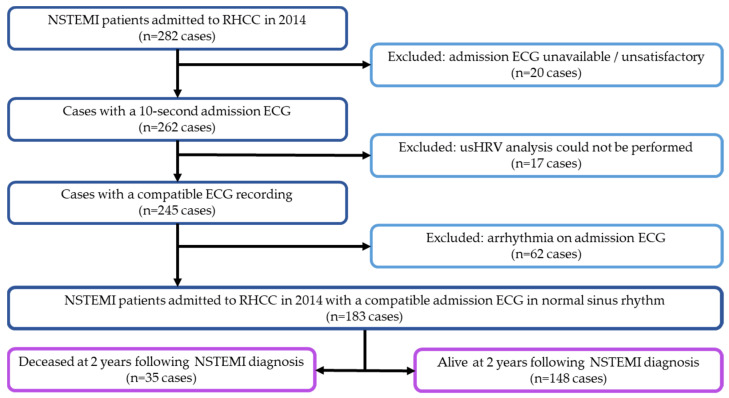
Flowchart of case selection and data availability. ECG—electrocardiogram; usHRV—ultra-short heart rate variability; NSTEMI—non-ST-elevation myocardial infarction; RHCC—Rambam Health Care Campus.

**Figure 2 jcm-13-07255-f002:**
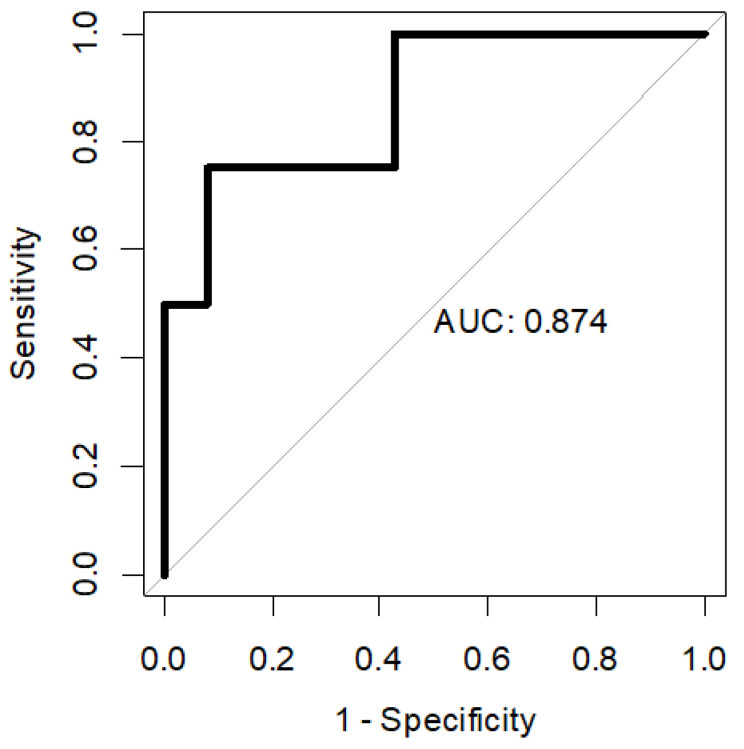
In-hospital VF or VT—receiver-operating characteristic (ROC) curve. AUC 95% CI: 0.674–1.000.

**Table 1 jcm-13-07255-t001:** Baseline characteristics of patients, grouped according to 2-year survival status.

Variable	Total (*n* = 183)	Alive (*n* = 148, 81%)	Deceased (*n* = 35, 19%)	*p*-Value
Age, years	67.1 ± 14.4	64.3 ± 13.7	79.0 ± 11.0	<0.0001 *
Sex, male	117 (63.9%)	95 (64.2%)	22 (62.9%)	0.781
Prior comorbidities				
Prior HF	12 (6.6%)	7 (4.7%)	5 (14.3%)	<0.0001 *
Prior SVT/AF/AFL	13 (7.1%)	9 (6.1%)	4 (11.4%)	0.001 *
Prior VT/VF	2 (1.1%)	2 (1.4%)	0 (0%)	0.172
Prior AVB	3 (1.2%)	1 (0.5%)	2 (8.8%)	<0.0001 *
Admission laboratory parameters				
MAP, mmHg	105.1 ± 19.2	104.9 ± 18.1	104.4 ± 24.1	0.758
HR, bpm	81.9 ± 16.8	80.2 ± 17.0	87.8 ± 14.4	0.014 *
O_2_ Saturation, %	95.1 ± 7.8	95.8 ± 4.4	91.6 ± 15.2	0.129
Hemoglobin, g/dL	12.9 ± 2	13.3 ± 1.6	11.1 ± 2.5	<0.0001 *
WBCs, ×10^3^/μL	9.4 ± 3.4	9.01 ± 2.72	10.8 ± 5.4	0.069
PLTs, ×10^3^/μL	224.6 ± 69.9	221 ± 61	233 ± 100	0.461
Glucose, mg/dL	164.3 ± 86.4	160 ± 83	183 ± 98	0.121
Creatinine, mg/dL	1.15 ± 1	1.03 ± 0.76	1.71 ± 1.57	0.016 *
BUN, mg/dL	21.6 ± 11.5	19 ± 9	31 ± 15	<0.0001 *
Potassium, mmol/L	4.1 ± 0.5	4.1 ± 0.5	4.3 ± 0.7	0.023 *
Sodium, mmol/L	137.4 ± 3.9	138 ± 4	136 ± 5	0.151
Troponin, ng/mL	1.19 ± 6.07	0.556 ± 1.515	4.369 ± 13.612	0.112
Admission HRV parameters (median ± IQR)				
SDNN, ms	12.44 ± 12.40	13.12 ± 13.23	10.28 ± 10.60	0.015 *
RMSSD, ms	14.62 ± 12.14	14.65 ± 11.85	13.43 ± 13.00	0.307
HTI	3.67 ± 1.70	3.67 ± 1.50	3.33 ± 1.58	0.204

*—*p*-value < 0.05. AF—atrial fibrillation; AFL—atrial flutter; AVB—atrioventricular block; BUN—blood urea nitrogen; HF—heart failure; HR—heart rate; HRV—heart rate variability; HTI—HRV triangular index; IQR—interquartile range; MAP—mean arterial pressure; PLTs—platelets; RMSSD—root mean square of successive differences; SDNN—standard deviation of NN intervals; SVT—supraventricular tachycardia; VF—ventricular fibrillation; VT—ventricular tachycardia; WBCs—white blood cells.

**Table 2 jcm-13-07255-t002:** Short- and long-term mortality—logistic regression.

Variable	30-Day Mortality		90-Day Mortality		1-Year Mortality		2-Year Mortality	
	OR (95% CI)	*p*-Value	OR (95% CI)	*p*-Value	OR (95% CI)	*p*-Value	OR (95% CI)	*p*-Value
Age, years	1.100 (1.019–1.187)	0.013 *	1.080 (1.029–1.133)	0.001 *	1.089 (1.049–1.131)	<0.0001 *	1.083 (1.049–1.119)	<0.0001 *
Sex, male	0.382 (0.074–1.961)	0.249	1.349 (0.356–5.115)	0.659	1.090 (0.428–2.772)	0.856	0.990 (0.438–2.239)	0.982
MAP, mmHg	0.967 (0.924–1.013)	0.165	0.968 (0.937–1.000)	0.052	0.989 (0.968–1.011)	0.363	0.999 (0.980–1.019)	0.976
HR, bpm	1.021 (0.977–1.066)	0.341	1.025 (0.994–1.057)	0.106	1.015 (0.991–1.039)	0.201	1.024 (1.003–1.046)	0.025 *
O_2_ Saturation, %	0.950 (0.845–1.069)	0.398	0.933 (0.861–1.011)	0.090	0.927 (0.859–1.001)	0.053	0.935 (0.868–1.008)	0.080
Hemoglobin, g/dL	0.838 (0.589–1.194)	0.330	0.761 (0.596–0.971)	0.028 *	0.583 (0.463–0.736)	<0.0001 *	0.609 (0.493–0.753)	<0.0001 *
WBCs, ×10^3^/μL	1.274 (1.078–1.506)	0.004 *	1.240 (1.089–1.413)	0.001 *	1.189 (1.068–1.325)	0.001 *	1.151 (1.042–1.272)	0.005 *
PLTs, ×10^3^/μL	0.979 (0.963–0.995)	0.012 *	0.997 (0.989–1.006)	0.602	1.002 (0.997–1.008)	0.323	1.002 (0.997–1.007)	0.400
Glucose, mg/dL	0.999 (0.989–1.009)	0.928	1.002 (0.996–1.008)	0.380	1.000 (0.995–1.005)	0.826	1.002 (0.998–1.006)	0.182
Creatinine, mg/dL	1.211 (0.741–1.981)	0.444	1.433 (1.034–1.986)	0.030 *	1.430 (1.029–1.988)	0.032 *	1.832 (1.109–3.026)	0.017 *
BUN, mg/dL	1.044 (0.997–1.095)	0.066	1.067 (1.028–1.107)	0.0005 *	1.071 (1.035–1.108)	<0.0001 *	1.079 (1.042–1.117)	<0.0001 *
Potassium, mmol/L	5.199 (1.644–16.437)	0.004 *	2.448 (1.000–5.993)	0.049 *	2.102 (1.028–4.298)	0.041 *	2.548 (1.304–4.978)	0.006 *
Sodium, mmol/L	0.897 (0.744–1.083)	0.261	0.893 (0.781–1.021)	0.099	0.960 (0.865–1.067)	0.457	0.926 (0.843–1.017)	0.108
Troponin, ng/mL	1.044 (0.989–1.103)	0.115	1.030 (0.977–1.086)	0.270	1.023 (0.974–1.076)	0.353	1.122 (0.973–1.293)	0.111
SDNN, ms	0.955 (0.860–1.061)	0.394	0.901 (0.809–1.004)	0.061	0.955 (0.907–1.005)	0.078	0.956 (0.916–0.999)	0.045 *
SDNN < 7.79 ms	0.586 (0.066–5.156)	0.631	1.966 (0.609–6.344)	0.258	1.391 (0.560–3.455)	0.476	1.479 (0.659–3.320)	0.342
RMSSD, ms	0.986 (0.933–1.042)	0.626	0.943 (0.868–1.024)	0.166	0.977 (0.944–1.012)	0.209	0.980 (0.954–1.008)	0.169
RMSSD < 10.22 ms	0.586 (0.066–5.156)	0.631	2.785 (0.885–8.767)	0.079	1.717 (0.706–4.172)	0.233	1.749 (0.789–3.878)	0.169
HTI	0.771 (0.383–1.552)	0.467	0.620 (0.357–1.078)	0.090	0.777 (0.549–1.099)	0.154	0.788 (0.583–1.065)	0.122

*—*p*-value < 0.05. BUN—blood urea nitrogen; HR—heart rate; HTI—HRV triangular index; MAP—mean arterial pressure; PLTs—platelets; RMSSD—root mean square of successive difference; SDNN—standard deviation of NN intervals; WBCs—white blood cells.

**Table 3 jcm-13-07255-t003:** In-hospital VF or VT—univariable logistic regression.

Variable	Odds Ratio (95% CI)	*p*-Value
Age, years	1.022 (0.953–1.096)	0.528
Sex, male	0.387 (0.053–2.826)	0.350
MAP, mmHg	0.947 (0.893–1.004)	0.072
HR, bpm	1.000 (0.942–1.061)	0.991
O_2_ Saturation, %	1.150 (0.676–1.955)	0.605
Hemoglobin, g/dL	0.869 (0.559–1.349)	0.532
WBCs, ×10^3^/μL	1.316 (1.081–1.602)	0.006 *
PLTs, ×10^3^/μL	0.996 (0.980–1.012)	0.633
Glucose, mg/dL	1.001 (0.991–1.012)	0.757
Creatinine, mg/dL	0.263 (0.007–8.873)	0.457
BUN, mg/dL	0.976 (0.875–1.089)	0.674
Potassium, mmol/L	0.861 (0.123–6.024)	0.881
Sodium, mmol/L	0.903 (0.719–1.135)	0.384
Troponin, ng/mL	1.055 (0.999–1.115)	0.052
SDNN, ms	1.025 (1.004–1.048)	0.020 *
SDNN < 7.79 ms	0.992 (0.100–9.783)	0.995
RMSSD, ms	1.018 (1.003–1.033)	0.016 *
RMSSD < 10.22 ms	0.992 (0.100–9.783)	0.995
HTI	1.149 (0.654–2.018)	0.628

*—*p*-value < 0.05. BUN—blood urea nitrogen; HR—heart rate; HTI—HRV triangular index; MAP—mean arterial pressure; PLTs—platelets; RMSSD—root mean square of successive differences; SDNN—standard deviation of NN intervals; WBCs—white blood cells.

**Table 4 jcm-13-07255-t004:** In-hospital VF or VT—multivariable logistic regression.

Variable	Adjusted Odds Ratio	95% Confidence Interval	*p*-Value
WBCs, ×10^3^/μL	1.433	1.116–1.842	0.004 *
SDNN, ms	1.033	1.007–1.060	0.011 *

*—*p*-value < 0.05. SDNN—standard deviation of NN intervals; WBCs—white blood cells. Intercept: Adjusted odds ratio—0.0001.

## Data Availability

The datasets analyzed within this current study are available in the FigShare repository, using the following hyperlink: https://doi.org/10.6084/m9.figshare.26905000.v1 (accessed on 2 September 2024).

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
