# Peer review of "Prognostic Value of Ultra-Short Heart Rate Variability Measures Obtained from Electrocardiogram Recordings of Hospitalized Patients Diagnosed with Non-ST-Elevation Myocardial Infarction"

_jcm, 2024, doi:10.3390/jcm13237255_

Round 1
Reviewer 1 Report
Comments and Suggestions for Authors
The prognostic assessment of the NSTEMI patient population has always been a focus of clinical attention. Due to the high heterogeneity in the prognosis of NSTEMI patients, existing risk scoring systems (such as TIMI and GRACE) have limited predictive capabilities, a clinical issue that urgently needs to be addressed. Heart rate variability (HRV) has a certain research foundation as a tool for predicting cardiovascular events, but its specific predictive value in NSTEMI patients remains unclear. This study included 183 NSTEMI patients hospitalized at Rambam Medical Center in 2014. By analyzing usHRV indicators such as SDNN and RMSSD obtained from 10-second ECG recordings, it was found that lower usHRV was significantly associated with 2-year mortality (with marginal significance at 90 days and 1 year), and higher usHRV was associated with an increased risk of in-hospital ventricular arrhythmias.
Before this manuscript meets the criteria, the following issues may need to be addressed:
1. The authors describe the conclusions of this study at the end of the introduction, which is clearly not in line with writing norms and needs correction.
2. The TIMI risk score is mainly used to predict the risk of major adverse cardiac events (such as death, myocardial infarction, or the need for urgent revascularization) in patients with non-ST-segment elevation myocardial infarction (NSTEMI) and unstable angina within 14 days. The GRACE risk score is used to assess the risk of death in patients with acute coronary syndrome during hospitalization and within 6 months to 1 year after discharge. These two scoring systems mainly focus on short-term and medium-term prognosis (from hospitalization to 6 months), while the usHRV discussed in this study not only predicts short-term prognosis but also shows predictive value for long-term prognosis (2 years), which is an important supplement to existing predictive models. Moreover, these traditional scoring systems require a comprehensive evaluation of multiple clinical indicators, whereas usHRV only requires a 10-second ECG recording, which is more convenient to operate. Therefore, the introduction section can elaborate more on the predictive endpoints and limitations of the TIMI and GRACE scores to better highlight the advantages of usHRV in long-term prognosis prediction. The authors are advised to reorganize the language and revise this part.
3. Regarding the sample size issue: Initially, 282 patients were included, and after exclusions, 183 patients were finally included. The incidence of primary endpoint events: 2-year mortality: 35 cases (19%), in-hospital heart failure: 39 cases (21.3%), ventricular tachycardia/ventricular fibrillation: 3 cases (1.6%), atrioventricular block: 9 cases (4.9%) (data obtained from the abstract). Due to the small number of individual events, especially only 3 cases of ventricular arrhythmia, the statistical power of the relevant analysis is severely insufficient. The current sample size is insufficient for the analysis of rare events. In multivariable analysis, according to statistical principles, at least 10 events are required for each predictor variable. Considering that the study involves multiple predictive factors, the current sample size may not be sufficient to support robust multivariable analysis. Therefore, the authors need to carefully consider the sample size issue and should explain the sample size calculation process in the methods section of the article. The authors should explicitly point out the limitations of insufficient sample size in the discussion section, particularly its impact on the analysis of rare events, the reliability of the multivariable analysis, and the limitations of the generalizability of the results.
4. In the results section, the numbers of in-hospital new-onset HF or exacerbation in preexisting HF, VF or VT, SVT, AF, or AFL are not clear enough, and the authors are advised to supplement this information.
5. The reviewers noticed that the authors used p-value<0.09 as a standard for trend significance in univariate analysis. Please explain the reason for choosing this standard.
6. In-hospital VF or VT – Receiver-operating characteristic (ROC) curve. AUC 95% CI: 0.674 - 1.000. Such a wide confidence interval usually indicates high uncertainty in the estimate. This may be due to insufficient sample size or too few events. In this case, with only 3 events occurring, it may lead to the width of the confidence interval. Additionally, the lower limit of the confidence interval at 0.674 indicates that the model's minimum discriminative ability is slightly higher than random guessing (AUC=0.5) but not very good. The upper limit of 1.000 indicates a possible perfect discriminative ability. However, such a perfect upper limit is usually unrealistic, especially when data is limited. In summary, the confidence interval for the AUC from 0.674 to 1.000 indicates a high degree of uncertainty in the results, and the authors are advised to interpret it cautiously.
7. A two-year survival analysis revealed that low SDNN significantly increases the risk of long-term mortality following NSTEMI (OR: 0.956, p-value: 0.045). An OR (odds ratio) of 0.956, close to 1, suggests that the impact of low SDNN on long-term mortality is relatively small. Furthermore, since the OR is close to 1, the result may be statistically significant but may have limited practical significance. The p-value is 0.045, slightly below the usual significance level of 0.05. This suggests that the result is statistically significant, but the borderline significance may be influenced by sample size, data variability, or other factors. Additionally, the confidence interval is missing here, which is an important indicator for assessing the uncertainty of the effect size estimate.
8. The discussion section lacks a discussion of the limitations of the above results. The manuscript lacks a limitations section.
9. In the manuscript, I noticed that some template instructions have not been removed, including the conclusion and patents sections. It is recommended to carefully check other parts of the manuscript to ensure that no similar template instructions are left.
Suggestion: The authors should make comprehensive revisions and additions based on the issues mentioned above to improve the quality and completeness of the manuscript.
Author Response
Comment 1: The authors describe the conclusions of this study at the end of the introduction, which is clearly not in line with writing norms and needs correction.
Response 1: We thank the reviewer for pointing out this issue. We agree with the comment and therefore, we removed the relevant content from the introduction section. The revised manuscript now contains only the hypothesis and its potential value (page 2, Introduction section paragraph 5, lines 84-88).
Comment 2: The TIMI risk score is mainly used to predict the risk of major adverse cardiac events (such as death, myocardial infarction, or the need for urgent revascularization) in patients with non-ST-segment elevation myocardial infarction (NSTEMI) and unstable angina within 14 days. The GRACE risk score is used to assess the risk of death in patients with acute coronary syndrome during hospitalization and within 6 months to 1 year after discharge. These two scoring systems mainly focus on short-term and medium-term prognosis (from hospitalization to 6 months), while the usHRV discussed in this study not only predicts short-term prognosis but also shows predictive value for long-term prognosis (2 years), which is an important supplement to existing predictive models. Moreover, these traditional scoring systems require a comprehensive evaluation of multiple clinical indicators, whereas usHRV only requires a 10-second ECG recording, which is more convenient to operate. Therefore, the introduction section can elaborate more on the predictive endpoints and limitations of the TIMI and GRACE scores to better highlight the advantages of usHRV in long-term prognosis prediction. The authors are advised to reorganize the language and revise this part.
Response 2: We thank the reviewer for this comment and the topics it discusses. The reviewer may find the revised paragraph in page 2, Introduction section paragraph 2, lines 45-55. We have taken these topics into consideration and elaborated on the endpoints of each risk score, as well as emphasized these points in the discussion section (page 9, Discussion section paragraph 5, lines 303-307). We revised the relevant part according to the reviewer’s request and hope that the current paragraph brings forward the topics that needed to be better discussed.
Comment 3: Regarding the sample size issue: Initially, 282 patients were included, and after exclusions, 183 patients were finally included. The incidence of primary endpoint events: 2-year mortality: 35 cases (19%), in-hospital heart failure: 39 cases (21.3%), ventricular tachycardia/ventricular fibrillation: 3 cases (1.6%), atrioventricular block: 9 cases (4.9%) (data obtained from the abstract). Due to the small number of individual events, especially only 3 cases of ventricular arrhythmia, the statistical power of the relevant analysis is severely insufficient. The current sample size is insufficient for the analysis of rare events. In multivariable analysis, according to statistical principles, at least 10 events are required for each predictor variable. Considering that the study involves multiple predictive factors, the current sample size may not be sufficient to support robust multivariable analysis. Therefore, the authors need to carefully consider the sample size issue and should explain the sample size calculation process in the methods section of the article. The authors should explicitly point out the limitations of insufficient sample size in the discussion section, particularly its impact on the analysis of rare events, the reliability of the multivariable analysis, and the limitations of the generalizability of the results.
Response 3: We agree with the reviewer that the study’s sample size was an important factor that influenced the statistical power of its results and the ability to draw valid conclusions from them. Our study included all patients diagnosed with NSTEMI at RHCC during 2014 for whom the ECG requirements were met. With that, several prior studies have dealt with mortality and severe short-term complications after NSTEMI, with a reported prevalence of 4%-7% (see references below). Given this reported prevalence, our sample size calculation for population proportion ranged from 164 to 278 patients. Nevertheless, the incidence of rare events in the cohort was in fact low and limited the validity of our results. The multivariable analysis of VF/VT was performed to provide the univariable analysis in a wider context and harder statistical examination. We agree with the reviewer that the wide CI limits the generalizability of this result and addressed this point in the discussion (page 9, Discussion section paragraph 4, lines 301-302).
A larger sample size, accompanied by a greater number of individual complications, would influence not only the results concerning VF/VT, but also those of other complications as well. We understand the reviewer’s reservations concerning the results and point to this issue both within the discussion (page 8, Discussion section paragraph 3, lines 273-276; page 9, Discussion section paragraph 3, lines 285-287) and as a limitation of the study. This was incorporated in a limitations section, as requested by the reviewers (page 9, Limitations section, lines 316-327).
- Guimarães, P.O.; Sampaio, M.C.; Malafaia, F.L.; Lopes, R.D.; Fanaroff, A.C.; de Barros e Silva, P.G.M.; dos Santos, T.M.; Okada, M.Y.; Mouallem, A.R.E.; Diniz, M.S.; Custódio, J.V.; Garcia, J.C.T.; Furlan, V. Clinical outcomes and need for intensive care after non-ST-segment-elevation myocardial infarction. Eur J Intern Med. 2020, 76, 58–63.
- Song, C.; Fu, R.; Li, S.; Yang, J.; Wang, Y.; Xu, H.; Gao, X.; Liu, J.; Liu, Q.; Wang, C.; Dou, K.; Yang, Y. Simple risk score based on the China Acute Myocardial Infarction registry for predicting in-hospital mortality among patients with non-ST-segment elevation myocardial infarction: Results of a prospective observational cohort study. BMJ Open. 2019, 9, 1-7.
- Vasquez-Rodriguez, J.F.; Idrovo-Turbay, C.; Perez-Fernandez, O.M.; Cruz-Tapias, P.; Isaza, N.; Navarro, A.; Medina-Mur, R.; Ramirez-Lovera, V.; Giraldo, L.E.; Ariza, N.; Carreno Jaimes, M.; Isaza, D. Risk of complications after a non-ST segment elevation acute myocardial infarction in a Latin-American cohort: An application of the ACTION ICU score. Heart Lung. 2023, 57, 124–129.
Comment 4: In the results section, the numbers of in-hospital new-onset HF or exacerbation in preexisting HF, VF or VT, SVT, AF, or AFL are not clear enough, and the authors are advised to supplement this information.
Response 4: We thank the reviewer for raising this point. In the revised manuscript we added the numbers of new-onset complications to the results section (page 6, Results section 3.3, lines 218-219; page 6, Results section 3.4, line 224; page 8, Results section 3.5, line 249; page 8, Results section 3.6, line 254).
Comment 5: The reviewers noticed that the authors used p-value<0.09 as a standard for trend significance in univariate analysis. Please explain the reason for choosing this standard.
Response 5: We understand the reviewers’ reservations concerning the use of p-value<0.09 to describe trend significance and added a justification for this threshold in the Materials and Methods section to reason the use of p-value<0.09 (page 4, Materials and Methods section paragraph 5, lines 151-154). In the manuscript we pointed to a trend significance only within the appropriate context of significant results and made sure not to draw conclusions that rely on such results (page 8, Discussion section paragraph 3, lines 265-266).
Comment 6: In-hospital VF or VT – Receiver-operating characteristic (ROC) curve. AUC 95% CI: 0.674 - 1.000. Such a wide confidence interval usually indicates high uncertainty in the estimate. This may be due to insufficient sample size or too few events. In this case, with only 3 events occurring, it may lead to the width of the confidence interval. Additionally, the lower limit of the confidence interval at 0.674 indicates that the model's minimum discriminative ability is slightly higher than random guessing (AUC=0.5) but not very good. The upper limit of 1.000 indicates a possible perfect discriminative ability. However, such a perfect upper limit is usually unrealistic, especially when data is limited. In summary, the confidence interval for the AUC from 0.674 to 1.000 indicates a high degree of uncertainty in the results, and the authors are advised to interpret it cautiously.
Response 6: We agree with the reviewer that the incidence of the secondary endpoint event of VF/VT is small and thus, limits the ability to interpret the results. As described in response 4, we referred to the wide CI in the discussion (page 8, Discussion section paragraph 3, lines 273-276) and limitations section (page 9, Limitations section, lines 317-320).
Comment 7: A two-year survival analysis revealed that low SDNN significantly increases the risk of long-term mortality following NSTEMI (OR: 0.956, p-value: 0.045). An OR (odds ratio) of 0.956, close to 1, suggests that the impact of low SDNN on long-term mortality is relatively small. Furthermore, since the OR is close to 1, the result may be statistically significant but may have limited practical significance. The p-value is 0.045, slightly below the usual significance level of 0.05. This suggests that the result is statistically significant, but the borderline significance may be influenced by sample size, data variability, or other factors. Additionally, the confidence interval is missing here, which is an important indicator for assessing the uncertainty of the effect size estimate.
Response 7: We thank the reviewer for raising this issue. We understand that an OR of 0.956 may signify a low impact of low SDNN on 2-year survival and addressed this concern in the discussion section (page 8, Discussion section paragraph 2, lines 267-269). We also added the 95% CI to the analysis results to clarify any uncertainties (page 6, Results section, Table 2, lines 205-209).
Comment 8: The discussion section lacks a discussion of the limitations of the above results. The manuscript lacks a limitations section.
Response 8: We greatly appreciate the point raised by the reviewer. We considered the previous comments made by the reviewer and added a limitations section to the manuscript, in which we addressed the abovementioned issues and additional limitations (page 9, Limitations section, lines 316-327).
Comment 9: In the manuscript, I noticed that some template instructions have not been removed, including the conclusion and patents sections. It is recommended to carefully check other parts of the manuscript to ensure that no similar template instructions are left.
Response 9: We apologize for our misuse of the journal's template. We have made sure to remove all the template instructions from the revised manuscript.

Reviewer 2 Report
Comments and Suggestions for Authors
The article provides a comprehensive analysis of the potential prognostic role of ultra-short HRV in NSTEMI patients. The study’s findings regarding the predictive value of low usHRV for long-term mortality and high usHRV for ventricular arrhythmias are noteworthy and could have significant clinical implications.
The article is well written and organized. I have only few suggestions.
1. Refine the introduction to focus more closely on ultra-short HRV measures' clinical potential in the NSTEMI context, and state specific hypotheses or expected outcomes.
2. While the methods are mostly clear, I suggest to expand the section from additional details on how ECG quality and recording conditions were standardized.
3. The discussion could benefit from a more explicit analysis of study limitations. For instance, the relatively small number of patients with ventricular arrhythmias limits the robustness of conclusions drawn about the association between high usHRV and arrhythmic risk.
4 In the discussion some interpretations of the data are speculative. For example, the hypothesized sino-atrial node dysfunction as a cause of high HRV could be better supported with citations or further context. This point could be flagged as a preliminary hypothesis that needs validation in future studies.
5. the discussion does not offer sufficient detail on future applications of usHRV as a clinical tool. Expanding on how usHRV could fit into clinical workflows, including integration with other risk scores or in real-time monitoring, would enhance the practical relevance of the research.
Tables, figures and english are fine.
Author Response
Comment 1: Refine the introduction to focus more closely on ultra-short HRV measures' clinical potential in the NSTEMI context, and state specific hypotheses or expected outcomes.
Response 1: We thank the reviewer for highlighting this need. Accordingly, we have refined the introduction to emphasize the clinical potential of ultra-short HRV measures as an easily applicable tool that may be particularly useful in guiding early treatment decisions. (page 2, Introduction section paragraph 4, lines 76-78). We also improved the description of our hypothesis (page 2, Introduction section paragraph 4, lines 81-83).
Comment 2: While the methods are mostly clear, I suggest to expand the section from additional details on how ECG quality and recording conditions were standardized.
Response 2: We thank the reviewer for this comment and agree on the importance of high-quality data and standardization methods. Therefore, we provided a more detailed description on how ECG recordings were obtained and analyzed to ensure standardized conditions (page 3, Materials and Methods section paragraph 4, lines 118-126). Additionally, we included more detailed information on the data collection process in the revised manuscript (page 3, Materials and Methods section paragraph 3, lines 107-112).
Comment 3: The discussion could benefit from a more explicit analysis of study limitations. For instance, the relatively small number of patients with ventricular arrhythmias limits the robustness of conclusions drawn about the association between high usHRV and arrhythmic risk.
Response 3: We thank the reviewer for pointing out this issue and agree that limitations should be discussed. As requested by reviewer 1, we added a limitations section to the revised manuscript, where we explicitly addressed the low number of rare outcome events as a limiting factor in our study (page 9, Limitations section, lines 316–327).
Comment 4: In the discussion some interpretations of the data are speculative. For example, the hypothesized sino-atrial node dysfunction as a cause of high HRV could be better supported with citations or further context. This point could be flagged as a preliminary hypothesis that needs validation in future studies.
Response 4: We agree with the reviewer on the topic discussed in this comment and understand the ideas presented in the discussion of high usHRV and SAN dysfunction should rely on more solid scientific evidence. The discussion section has been thoroughly revised to present these ideas from a more conservative perspective. Specifically, we rephrased the section discussing SAN dysfunction and used more careful wording to reflect the need for further validation of this preliminary hypothesis (page 9, Discussion section paragraph 4, lines 293-302).
Comment 5: The discussion does not offer sufficient detail on future applications of usHRV as a clinical tool. Expanding on how usHRV could fit into clinical workflows, including integration with other risk scores or in real-time monitoring, would enhance the practical relevance of the research.
Response 5: We appreciate the reviewer’s insightful comment and fully agree with the issues raised. In response, we expanded the discussion section to describe how usHRV measurement could be applied within the clinical workflow. The revised manuscript now emphasizes the potential advantages of usHRV for rapidly identifying high-risk patients and predicting long-term outcomes following NSTEMI. However, we also recognize that additional validation is required before these applications can be broadly implemented. (page 9, Discussion section paragraph 5, lines 305-313).

Round 2
Reviewer 1 Report
Comments and Suggestions for Authors
Based on my review of the authors' responses and modifications, they have made a commendable effort to address most of the reviewer's comments and concerns appropriately and comprehensively. However, there are several areas that may require further attention:
1. Use of p<0.09 as a threshold for trend significance: This issue needs further attention for the following reasons: While the authors provide some explanation in the Materials and Methods section (paragraph 5, lines 151-154), this may not sufficiently justify the use of this non-standard threshold. Furthermore, using a more lenient significance criterion might lead to overinterpretation of what could be mere random variations as "trends." Additionally, using a non-standard significance level may make it difficult to compare the study's findings with other research using the standard p<0.05 threshold. The reviewer suggests a stronger scientific justification for using p<0.09 or reverting to the more traditional p<0.05 standard while discussing results with p-values between 0.05 and 0.09 as "possible trends" requiring further investigation.
2. The reviewer suggests that the authors provide a more in-depth discussion of the clinical significance of results with small effect sizes (such as the relationship between SDNN and mortality) in the discussion section.
3. The reviewer recommends expanding the limitations section to include a more comprehensive discussion of potential limitations related to sample characteristics, study design, and statistical analysis methods. Additionally, the last sentence of the limitations section is incomplete. This is a serious writing or editing error that needs immediate correction.
4. The reviewer noticed two tables labeled "Table 3" in the manuscript. The authors should verify whether this is a writing or editing error. Thorough proofreading of the entire manuscript is necessary to ensure there are no spelling mistakes, grammatical issues, or unclear expressions.
Author Response
Comment 1: Use of p<0.09 as a threshold for trend significance: This issue needs further attention for the following reasons: While the authors provide some explanation in the Materials and Methods section (paragraph 5, lines 151-154), this may not sufficiently justify the use of this non-standard threshold. Furthermore, using a more lenient significance criterion might lead to overinterpretation of what could be mere random variations as "trends." Additionally, using a non-standard significance level may make it difficult to compare the study's findings with other research using the standard p<0.05 threshold. The reviewer suggests a stronger scientific justification for using p<0.09 or reverting to the more traditional p<0.05 standard while discussing results with p-values between 0.05 and 0.09 as "possible trends" requiring further investigation.
Response 1: We appreciate the reviewer’s thoughtful feedback regarding the use of p<0.09 as a threshold for non-significant trends. After careful consideration of the comment, we acknowledge the importance of adhering to widely accepted standards of statistical significance to ensure clarity and comparability across studies. Consequently, we have revised the manuscript to remove references to non-significant trends based on p<0.09.
Comment 2: The reviewer suggests that the authors provide a more in-depth discussion of the clinical significance of results with small effect sizes (such as the relationship between SDNN and mortality) in the discussion section.
Response 2: We appreciate the reviewer’s valuable suggestion to discuss the clinical significance of results with small effect sizes, particularly the relationship between SDNN and mortality. In response, we have edited and expanded the Discussion section to address this point (page 8, Discussion section paragraph 2, lines 263-270).
Our results indicate that for every 1 millisecond increase in SDNN, the risk of mortality is reduced by 4.4%. This reduction, while seemingly small, is clinically meaningful given the cumulative impact of even modest improvements in autonomic regulation on patient outcomes. However, we acknowledge that the OR of 0.956, with a 95% CI approaching 1, suggests some uncertainty in the precise effect size, highlighting that the observed association could be slightly weaker or stronger. While the effect size may not be large, its potential clinical utility lies in its integration with other risk predictors, contributing to a more nuanced understanding of patient prognosis. This is especially valuable in acute care settings, where quick and reliable indicators are essential for risk stratification.
We have incorporated these considerations into the Discussion section to provide a balanced interpretation of our findings and their implications for clinical practice. Thank you for highlighting this important aspect of our work.
Comment 3: The reviewer recommends expanding the limitations section to include a more comprehensive discussion of potential limitations related to sample characteristics, study design, and statistical analysis methods. Additionally, the last sentence of the limitations section is incomplete. This is a serious writing or editing error that needs immediate correction.
Response 3: We thank the reviewer for highlighting the need for a more comprehensive discussion in the limitations section. In response, we have expanded the section to address the constraints of our study design, specifically the retrospective nature, which relies on pre-existing data prone to incompleteness and bias, and the single-center setting, which limits generalizability due to potential selection bias and institution-specific practice patterns (page 9, Limitations section paragraph 1, lines 327-332). Additionally, we corrected the incomplete sentence in the limitations section to ensure clarity and accuracy.
Comment 4: The reviewer noticed two tables labeled "Table 3" in the manuscript. The authors should verify whether this is a writing or editing error. Thorough proofreading of the entire manuscript is necessary to ensure there are no spelling mistakes, grammatical issues, or unclear expressions.
Response 4: We thank the reviewer for this comment. We carefully proofread the manuscript and made sure that all tables and figures are properly numbered and are found in the manuscript in the correct order with the appropriate references in the text. As requested, we can verify that “Table 3” describes the results of the univariable analysis for in-hospital VF or VT (page 7, Results section 3.4., line 233), and the corrected “Table 4” describes the multivariable analysis for this outcome event (page 7, Results section 3.4., line 238).